# Quality of referral system and associated factors among referred clients referred to dessie comprehensive specialized hospital, Northeast, Ethiopia

Biruk Abera[1], Toyib Yasin[2], Hiwot Gizaw[3], Yonas Fissha Adem [4]*

1 Department of Midwife, Dessie Comprehensive Specialized Hospital, Dessie, Ethiopia, 2 Department of Health Systems and Management, Wollo University, Dessie, Ethiopia, 3 Department of Health Informatics, Wollo University, Dessie, Ethiopia, 4 Department of Public Health, Dessie College of Health Sciences, Dessie, Ethiopia

* yonasfissha029@gmail.com

## Abstract

### Background

An effective referral system acts as a connection among the three tiers of healthcare in Ethiopia, ensuring the seamless provision of appropriate health services. The success of the referral system is a key factor influencing the overall quality and strength of healthcare delivery. Therefore, this study aimed to assess the quality of the referral system and associated factors among referred clients at Dessie Comprehensive Specialized Hospital, Northeast Ethiopia.

### Methods

A facility-based cross-sectional study was conducted on 413 referred patients at Dessie Comprehensive Specialized Hospital from January 15, 2023, to February 15, 2023. The study participants were selected using a systematic random sampling technique. Data entry and processing were performed using EpiData version 4.4.2 and exported to SPSS version 26.0 for further analysis. Both bivariate and multivariable binary logistic regression analyses were used to identify factors associated with the quality of the referral system. Crude odds ratios (COR) and adjusted odds ratios (AOR), along with 95% confidence intervals (CI), were calculated. A p-value < 0.05 in the final model was considered statistically significant. This study assessed the referral system's structure (forms, transport registers, focal persons), process (14 referral paper components like patient details and reasons), and outcomes (patient satisfaction surveys).

**Data availability statement:** All relevant data are within the manuscript and its Supporting Information files.

**Funding:** The author(s) received no specific funding for this work.

**Competing interests:** The authors have declared that no competing interests exist.

## Result

The overall quality of the referral system to Dessie Comprehensive Specialized Hospital was 62.5% (95% CI: 57.6%–66.8%). The overall input quality of the referral system was 90.3% (95% CI: 87.7%–93.2%), and the overall process quality was 88.3% (95% CI: 86.5%–91.6%). Patient satisfaction was 44.6% (95% CI: 40.0%–49.2%). Transport mode was significantly associated with the quality of the referral system: patients transported by ambulance [AOR: 6.59, 95% CI: 3.49–12.43], by vehicle [AOR: 5.62, 95% CI: 2.33–10.52], by public taxi [AOR: 3.82, 95% CI: 1.25–8.65], and those referred from rural areas [AOR: 0.33, 95% CI: 0.19–0.56].

## Conclusion

Generally, a significant proportion of referred clients received poor referral quality. Improvements are needed by establishing and facilitating a transportation mechanism and a feedback system.

## Introduction

Referral is a two-way process and ensures that a continuum of care is maintained for patients or clients [1]. Referral refers to the process by which professionals and institutions communicate and collaborate to protect, promote, and restore an individual's health. The movement of a patient to another level of care may be internal, upward, downward, or lateral, ensuring continuity of care [2]. Referrals are the link and interface between healthcare providers in primary and specialty care settings [3]. Achieving the highest standard of health is a fundamental human right that can be supported by an effective referral system, which enhances continuity of care across different levels [4]. Even though nearly all low-income countries have weak health referral systems and inefficient services, different levels of care affect the overall outcome of the health system, which in turn influences health outcomes [5].

Quality has three principal components: structure, process, and outcome. Structure refers to the attributes of the settings where care is provided, including resources, staff, equipment, referral forms, transportation, and the presence of a designated focal person—a trained individual responsible for coordinating and managing referral activities within the health facility. Process encompasses every aspect of delivering healthcare and pertains to interactions between clinicians and patients, as well as the quality of referral and feedback documents. Outcome focuses on the results or effects of the care provided, such as patient satisfaction [6,7]. Evidence shows that even when all the necessary structural components are available, the quality of care may still be poor unless there is appropriate use of available resources to ensure effective case management [6].

Referral systems are designed to reduce wait times while also enhancing workflow effectiveness [8]. An effective referral system ensures close collaboration among all levels of healthcare, enabling individuals to receive the best possible care [9]. By

optimizing the use of healthcare facilities, a strong referral system enhances the overall effectiveness of the healthcare system. It strengthens outlying medical institutions and improves the ability of professionals at the lower levels of the referral network to make informed decisions [1].

A study conducted by the Archives of Internal Medicine indicated that more than 105 million clients visit specialists by referral, of whom half complete the course of services. Less than 25% of clients complete the services as intended by the referred healthcare providers [10]. Globally, approximately 2.8 million neonates and 295,000 mothers die each year. Almost two-thirds of these deaths occur in Africa and Asia [11]. In Africa, a research report from Zimbabwe suggested that more than half of the clients were treated at health centers, even when their problems required referral to a higher-level hospital [12]. In low-income countries, the rates of seeking care at a hospital referred from primary health care were 7% in Nigeria, 27% in Kenya, and 38% in Zimbabwe [12]. The increase in deaths among the communities was related to ineffective patient referrals [11]. Many countries have introduced the use of an electronic referral system, which has recently gained considerable attention for its importance in replacing the manual referral process and reducing the time required to find appropriate specialists [13,14]. However, Ethiopia hasn't yet adopted the e-referral system. A study of the patient electronic referral system has been conducted by [15] and concluded that healthcare organizations are highly recommended to implement the electronic medical referral system.

Despite the significant expansion in the number of health posts and health centers in Ethiopia, people routinely accessed hospitals without a formal referral from a health center or health post and without seeking any prior source of care [8,16]. Ethiopia's primary care system has a weak referral process for severely ill patients, including children [17]. Inadequate access to quality care is one of the factors that can increase the risk of maternal and neonatal death. Despite significant improvements in health status, maternal mortality remains high in the majority of reported deaths from the Tigray region [16].

To improve the referral system, the Ethiopian FMOH is promoting accelerated road construction, effective ambulance services, and efficient communication systems between referring and referral units [18]. Increasing people's access to healthcare and reducing unnecessary costs are the objectives of health referral care plans [19]. There are different kinds of literature and studies, but no such research has been conducted in this study area. Additionally, the quality of the referral system may vary from place to place. Therefore, this study aimed to assess the quality of the referral system and associated factors at the local level, which are important for developing strategies and designing appropriate interventions.

## Methods and materials

### Study design, area, and period

A facility-based cross-sectional study was conducted at Dessie Comprehensive Specialized Hospital from January 15, 2023, to February 15, 2023. Dessie City is the capital of the South Wollo Zone in the Amhara Region, located 401 kilometers from Addis Ababa, the capital city of Ethiopia, and 480 kilometers from Bahir Dar. The hospital was established in 1942 at Kurkur. Later, in 1962, it was named Asfaw Wossen and located at the existing site. In 2005, the name was changed to Dessie Referral Hospital, and in 2020, it was upgraded to Dessie Comprehensive Specialized Hospital. Currently, it has over 303 administrative staff and more than 562 healthcare workers. In total, the hospital employs over 865 staff members. It also provides preventive, curative, and rehabilitative services to over 8 million people [20].

### Population

**Source population.** All patients who were referred to Dessie Comprehensive Specialized Hospital.
**Study population.** Those referred patients who had referral papers and arrived at Dessie Comprehensive Specialized Hospital during the study period.

## Eligibility Criteria

**Inclusion criteria.**  All referred patients to Dessie Comprehensive Specialized Hospital whose ages were 18 and above during the study period.

**Exclusion criteria.**  Clients who unable to communicate, and died on arrival were excluded.

**Sample size determination.**  The sample size (n) required for the study was calculated using the formula to estimate a single population proportion.

$$n = \frac{[(Z\alpha/2)^2\, P\,(1-P)]}{d^2}$$

With the assumptions of 95% Confidence level, 5% desired precision,

n = required sample size

Zα/2 = critical value for normal distribution at 95% confidence level which equals 1.96 (Z value at alpha = 0.05).

P = Proportion of good quality of the referral system which gives the larger sample size is 47% taken from the previous study (16).

d = an absolute precision (5% margin of error)

$$n = (1.96(1.96)\,(0.47)\,(0.53))/\,((0.05)2) = 383$$

Considering a non-response rate of 10% (which equals 38), the total sample size was 421 referred patients to Dessie Comprehensive Specialized Hospital.

## Sampling procedures and technique

To achieve the desired sample size of 421, a systematic random sampling technique was implemented at Dessie Comprehensive Specialized Hospital, where patient charts were organized based on their assigned order, facilitating straightforward selection regardless of patient arrival times(at night or any time). The number of referred patients was determined from the client flow rate during the data collection period, with the most recent hospital report indicating a total of 6,021 referred patients over the past three months, averaging approximately 2,007 patients per month. This led to the calculation of 'k' as 2007/421 ≈ 4.7, which was then rounded to 4. The selection process began with the first client chosen by simple random sampling among the first four referred patients. Subsequently, every fourth client was included in the study until the required sample size was reached. When a patient was unavailable at the scheduled sampling time, unable to communicate, or if the referral paper was lost, the next random sample was taken, and the data collection was conducted during the day. Additionally, contingency plans were in place so that if a research staff member fell ill or absence, other data collectors would be deployed to ensure continuous data collection.

## Study variables

**Dependent variable.**  Patient satisfaction (Good/poor)

**Independent variables.  Socio-demographic factors:** age, sex, marital status, educational status, residence, monthly income, and Number of family

**Case-related factors:** medical, Maternal and Child Health (MCH), surgical, and accident

**Facility-based factors:** health centers, private clinics, government and private hospitals

## Operational definitions

**Structure quality**:- Measurement was based on the availability of referral forms, transportation referral registration books, and focal persons. They were assessed as "Yes" if available and "No" if absent. The percentage was then computed. Those who scored 75% or above were considered good, while those who scored below 75% were considered poor [21].

**Process quality**: -Referral process (quality of referral and feedback paper)
The quality of the referral paper was assessed based on 14 components. Each component was recorded as "yes" if present and "no" if absent. A total score of 0–9 was considered poor in quality, while a score of 10–14 was considered good in quality [21].

The quality of the referral paper was assessed based on 14 components. Each component was recorded as "yes" if present and "no" if absent. A total score of 0–9 was considered poor in quality, while a score of 10–14 was considered good in quality [21].

**Total Process quality**:- Measured by the added components of the quality of the referral paper and components of the quality of the feedback paper, then percentages were computed. Those who scored 75% or above were considered good, and those who scored below 75% were considered poor [21].

**Outcome quality**: The total satisfaction score was calculated, and those who scored above the average were considered satisfied, while those who scored below the average were considered dissatisfied [21].

**Overall quality of referral system**:- Computed by three components of quality [22].

**Good quality of referral system**: The three measures of quality (structure, process, and outcome) are 75% or above [22].

**Poor quality of referral system**:- The sum of three measures of quality (structure, process, and outcome <75% [22].

**Referral Registry form**:-  At the facility level that properly documents outgoing and incoming referrals.

## Data collection procedure and quality control

The data were collected using structured, pre-tested, interviewer-administered questionnaires. The sections on structure (inputs), process, and outcomes of the questionnaire were developed based on Ethiopian guidelines for referral systems and relevant literature [23]. The assessment of structure quality focused on the availability of essential resources such as referral forms, transportation referral registration books, and designated focal persons. Process quality was evaluated based on the completeness of referral papers, which included 14 key components such as patient details, reason for referral, and referral date. Outcome quality was measured through patient satisfaction surveys conducted post-consultation, capturing insights into the patient's experience with the referral process. The questionnaires were translated into Amharic and back into English to ensure consistency. Data collection was carried out by four BSc nurses selected from other health facilities, under the supervision of two BSc nurses. Quality was maintained through the design of appropriate data collection tools, pre-testing, and continuous supervision. Before the actual data collection, training was provided to health extension worker data collectors for two days on data collection techniques to familiarize them with the tools. Participants were interviewed after giving informed consent. During the data collection process, the completed questionnaires were checked for completeness by the principal investigator.

## Data processing and analysis

Data were checked for completeness and consistency, then coded and entered into EPI-data version 4.4.2 and exported to SPSS version 26.0 for processing and analysis. Frequency distribution tables, graphs, and descriptive summaries were used to describe the study variables. Model fitness was assessed using the Hosmer-Lemeshow goodness-of-fit test. Bivariate and multivariable analyses were conducted to examine the association between the quality of the referral system and selected independent variables. The bivariate logistic regression model was used to identify candidate variables for the multivariable analysis. Variables with a significant bivariate test (p-value < 0.25) were included in the multivariable logistic regression model. Factors with a p-value less than 0.05, along with their adjusted odds ratio (AOR) and 95% confidence interval (CI), were considered statistically significant in their association with the quality of the referral system.

### Research ethics approval

The study was conducted after obtaining ethical approval from Wollo University, College of Medicine and Health Sciences, School of Public Health, with a reference number of CMH/1794/20/23. A formal letter from the College of Medicine and Health Sciences at Wollo University was submitted to Dessie Comprehensive Specialization Hospital to request their cooperation. The rights and dignity of respondents were also respected. Verbal informed consent was obtained from study participants to confirm their willingness to participate after explaining the purpose of the study. All information collected from participants was kept strictly confidential, and their names were not included in the questionnaire. All methods were performed in accordance with the Declaration of Helsinki.

## Results

### Socio-demographic characteristics of the study participants

A total of 421 referred cases were included in the study, resulting in a response rate of 413 (97.87%). Of these participants, 264 (63.9%) were from urban areas, while the remaining were from rural areas. Among the respondents, 212 (51.3%) were females. Of the 413 study subjects, 261 (63.2%) were married, and 196 (47.5%) identified as Orthodox by religion. Regarding educational status, 113 (27.4%) respondents had completed college or higher education, and 129 (31.2%) were government employees. Two hundred seventy-nine (67.6%) of the respondents had fewer than five family members. The mean monthly income of the participants was 3,961 Birr. (Table 1)

### Referring facilities

Of all the 413 study participants, nearly half (46.5%) were referred from health centers, 166 (40.2%) from government hospitals, 30 (7.3%) from private hospitals, 23 (5.6%) from private clinics, and only 2 (0.5%) from other health facilities.

### Referral cases

Of all the 413 study participants, 78 (18.9%) were referred for an accident, 151 (36.6%) for medical cases, 64 (15.5%) for surgical care, 109 (26.4%) for MCH, and 11 (2.7%) for other cases.

### Structural (input) part of the referral system

Concerning the structural components or availability of inputs for the referral system, 386 (93.5%) patients were referred from health facilities that had a referral focal person, 368 (89.1%) from facilities with a referral registry book, 374 (90.6%) from those with standard referral papers, and 352 (85.2%) from facilities that had transport for referred cases. The overall structural input quality of the referral system was 90.3% (CI 87.7–93.2%).

### Process of the referral system

Regarding the referral paper, the most frequently recorded component was the name of the health facility, while the least was laboratory investigation, which was recorded in 98.5% and 74.3% of referred patients, respectively. The details of the other components are shown in the following table. The overall quality of the referral paper was good in 391 (94.7%) (CI 92.5–96.6) of the study participants and poor in 22 (5.3%) (CI 3.4–7.5) of them. (Table 2)

Regarding the feedback paper, 371 (89.8%) of the total study participants received referral feedback, with the date of feedback being the most frequently recorded component. The date was recorded in 88.4% of the feedback provided. The last recorded component of the referral feedback was the appointment, with a frequency of 67.1%. The details of the other components are shown in the following table. (Table 3)

In this study, of those referral papers, 94.7% were scored as good in quality, whereas only 5.3% were poor in quality. The scores of the components of the feedback sent were good in 87.3% of the study subjects. The overall quality of the

**Table 1. Socio-demographic characteristics of study participants on Quality of referral system and associated factors among clients referred to Dessie Comprehensive Specialized Hospital, Northeast, Ethiopia, 2022/23(n = 413).**

| Variable | Frequency | Percent |
|---|---|---|
| Sex | | |
| Male | 201 | 48.7 |
| Female | 212 | 51.3 |
| Age (in years) | | |
| 18–24 years | 57 | 13.8 |
| 25–34 years | 144 | 39.4 |
| 35–44 years | 103 | 24.9 |
| 45 and above | 109 | 26.4 |
| Marital status | | |
| Single | 85 | 20.6 |
| Married | 261 | 63.2 |
| Divorced | 37 | 9.0 |
| Widowed | 30 | 7.3 |
| Religion | | |
| Orthodox | 196 | 47.5 |
| Muslim | 183 | 44.3 |
| Protestant | 23 | 5.6 |
| Catholic | 11 | 2.7 |
| Educational status | | |
| Unable to write and read | 40 | 9.4 |
| Read and write | 72 | 17.4 |
| Elementary | 89 | 21.5 |
| Secondary school completed | 99 | 24.0 |
| College and above completed | 113 | 27.4 |
| Occupational status | | |
| Farmer | 75 | 18.2 |
| House wife | 73 | 17.7 |
| Merchant | 69 | 16.7 |
| Government employee | 129 | 31.2 |
| Daily laborer | 27 | 6.5 |
| Student | 40 | 7.02 |
| Family size | | |
| Less than 5 | 279 | 67.6 |
| 5 and above | 134 | 32.4 |
| Place of residence | | |
| Urban | 264 | 63.9 |
| Rural | 149 | 36.1 |
| Monthly income(in ETB) | | |
| ≤ 1000 | 68 | 16.5 |
| 1001-2000 | 77 | 18.6 |
| >2000 | 268 | 64.9 |

**Table 2. Frequency distribution of components of the referral papers among referral cases (study subjects) of Dessie Comprehensive Specialized Hospital, Northeast, Ethiopia, 2022/23 (n = 413).**

| Component of the referral paper | Yes Frequency (%) | No Frequency (%) |
|---|---|---|
| Name of health facility | 407(98.5) | 6(1.5) |
| MRN | 386(93.5) | 27(6.5) |
| Age of the patient | 374(90.6) | 39(9.4) |
| Sex of a patient | 380(92.0) | 33(8.0) |
| Chief complaint | 399(96.6) | 14(3.4) |
| Duration of chief complaint | 334(80.9) | 79(19.1) |
| History of present illness | 345(83.5) | 68(16.5) |
| Physical examination | 344(83.3) | 69(16.7) |
| Laboratory investigation | 307(74.3) | 106(25.7) |
| Diagnosis | 365(88.4) | 48(11.6) |
| Management | 344(83.3) | 69(16.7) |
| Reason for referral | 393(95.2) | 20(4.8) |
| Name of referring professional | 395(95.6) | 18(4.4) |
| Date of referral | 397(96.1) | 16(3.9) |

referral feedback was good in 362 (87.7%) (CI 84.55–90.6) of the study participants and poor in 51 (12.3%) (CI 9.4–15.5) of them.

Over-process quality was good in 365 (88.3%) (CI 86.5%–91.6%) of study participants. (Table 4)

## Patient satisfaction

In this study, 282 (68.3%) of the referred patients agreed that health professionals explained the referral in a way they could understand. Additionally, 258 (62.5%) agreed that health professionals included them in deciding the reason for the referral. Conversely, 4 (1.0%) of the participants strongly disagreed that they were satisfied with the overall referral service, and 6 (1.5%) strongly disagreed that health professionals explained the referral in a comprehensible manner. The overall patient satisfaction with the referral system at Dessie Comprehensive Specialized Hospital was found to be 44.6% (95% CI: 40.0%–49.2%). (Table 5)

## Overall quality towards referral of health facilities

The overall quality of the referral system in this study was computed by considering the three components of quality measures (input, process, and outcome). Based on this, the overall quality of the referral system was 62.5% (95% CI: 57.6%–66.8%), while the remaining 37.5% was considered poor in quality. (Fig 1)

## Factors associated with patient satisfaction toward referral system

Both bivariate and multivariable logistic regression analyses were performed to identify the predictors of the quality of the referral system. In the bivariate analysis, sex, religion, marital status, educational status, occupation, residency, family size, mode of transportation, and type of case were found to be associated variables at P ≤ 0.25 and were subsequently entered into the multivariable logistic regression analysis.

In the final multivariable logistic regression analysis, mode of transportation and residency were significantly associated with the quality of the referral system at P ≤ 0.05(Table 6).

As the mode of referral transportation was significantly associated with patient satisfaction regarding referral to health facilities, those transported by ambulance were 6.59 times more likely to be satisfied than those transported on foot [AOR:

 

**Table 3. Frequency distribution of components of feedback paper among referral cases (study subjects) of Dessie Comprehensive Specialized Hospital, Northeast, Ethiopia, 2022/23 (n = 413).**

| Component of the feedback paper | Yes Frequency (%) | No Frequency (%) |
|---|---|---|
| Is there referral feedback? | 371(89.8) | 42(10.2) |
| Name Health facility | 339(82.1) | 74(17.9) |
| Diagnosis at the health facility | 325(78.7) | 88(21.3) |
| An investigation was done at the hospital | 345(83.5) | 68(16.5) |
| Diagnosis of the patient at the hospital | 355(86.0) | 58(14.0) |
| Hospitals Plan of Management | 346(83.8) | 67(16.2) |
| Appointment | 277(67.1) | 136(32.9) |
| Patient attending health professional | 362(87.7) | 51(12.3) |
| Date of feedback | 365(88.4) | 48(11.6) |

**Table 4. Assessment of referral and feedback paper quality among referral cases (study subjects) of Dessie Comprehensive Specialized Hospital, Northeast, Ethiopia, 2022/23 (n = 413).**

| Quality of referral and feedback paper | Frequency (%) |
|---|---|
| Referral paper (n = 413) | |
| 0–9(poor) | 22(5.3) |
| 10–14(good) | 391(94.7) |
| Feedback(n = 371) | |
| 0–6(poor) | 51(12.3) |
| 7–9(good) | 362(87.7) |
| Total process quality (413) | |
| Good | 365 (88.3%) |
| Poor | 48(11.7%) |

**Table 5. Patient satisfaction toward referral among referral cases (study subjects) of Dessie Comprehensive Specialized Hospital, Northeast, Ethiopia, 2022/23 (n = 413).**

| Satisfaction questions | Strongly disagree N (%) | Disagree N (%) | Neutral N (%) | Agree N (%) | Strongly agree N (%) |
|---|---|---|---|---|---|
| They referred me because they couldn't treat me at the health facility | 20(4.8) | 48(11.6) | 13(3.1) | 214(51.8) | 118(28.6) |
| Health professionals in health facilities call an ambulance if a client is very sick | 13(3.1) | 127(30.8) | 17(4.1) | 173(41.9) | 83(20.1) |
| Health professionals in health facilities ask patients to return to see how they are doing | 24(5.8) | 194(47.0) | 38(9.2) | 115(27.8) | 42(10.2) |
| When I'm sick I did not visit a traditional healer before I came to the health | 39(9.4) | 209(50.6) | 21(5.1) | 129(31.2) | 15(3.6) |
| There was difficulty in the referral | 25 (6.1) | 167(40.4) | 52(12.6) | 143(34.6) | 26(6.3) |
| I am satisfied with the overall referral service | 4(1.0) | 70(16.9) | 41(9.9) | 232(56.2) | 66(16.0) |
| Health professionals told me about the referral in a way I can understand | 6(1.5) | 43(10.4) | 25(6.1) | 282(68.3) | 57(13.8) |
| Health professionals include me in deciding about the reason for referral | 7(1.7) | 52(12.6) | 32(7.7) | 258(62.5) | 64(15.5) |
| Overall patient satisfaction | 44.6% (CI 40.00–49.200) | | | | |

6.59, CI (3.49–12.43)]. Similarly, individuals transported by private vehicle were 5.62 times more likely to be satisfied than those transported on foot [AOR: 5.62, CI (2.33–10.52)]. Additionally, those transported by other means, such as a public taxi, were approximately 3.82 times more likely to be satisfied than those transported on foot [AOR: 3.82, CI (1.25–8.65)].

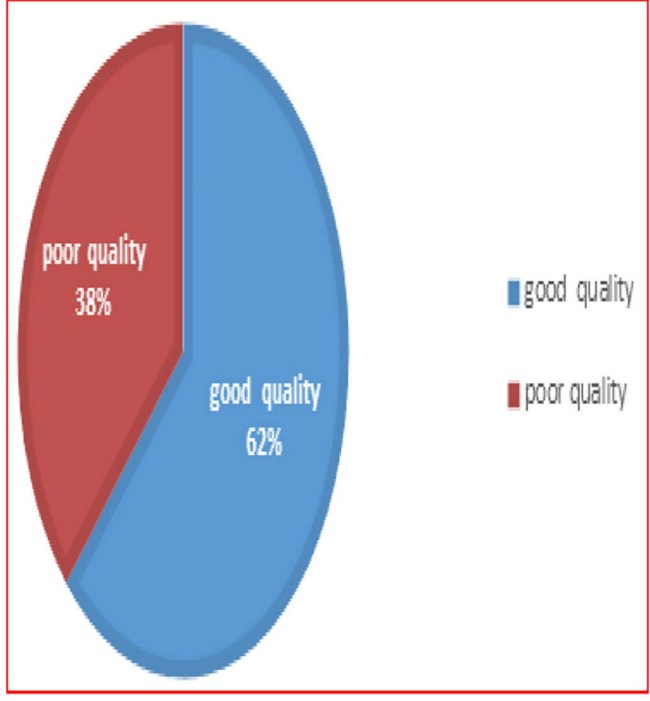

**Fig 1. Overall quality towards referral among referral cases (study subjects) of Dessie Comprehensive Specialized Hospital, Northeast, Ethiopia, 2022/2023(n = 413).**

The second independent variable that has been statistically associated with the level of patient satisfaction regarding referral to health facilities is residence, with those referred from rural areas being 67% less satisfied than those referred from urban areas [AOR = 0.33, 95% CI (0.19–0.56)].

## Discussion

This study aimed to assess the quality of the referral system and its associated factors at Dessie Comprehensive Specialized Hospital by measuring the three quality components—structure/input, process, and outcome—based on the Donabedian model of quality measurement. These three components are interconnected; a defect in one can affect the others. The overall structural quality of the system was 90.3% (CI 87.7–93.2%). This finding is higher than that of a study conducted in the Eastern zone of Tigray, where the structural quality of the referral system was 47% [18]. Similarly, the availability of inputs in this study was higher as compared to the study done in Malawi 58% [5]. The finding was also better as compared to the studies done in Iraq (21.5%) [18,21]. The difference could be due to variations in the study area and study period. Additionally, the discrepancy might stem from socioeconomic differences or the level of attention given to the structural aspects of the referral system in our country. Regarding the quality of referral papers, our study showed that 94.7% of them were of good quality. This indicates that 5.3% of referred patients had poor-quality referral papers, suggesting that patients were referred with insufficient details. Such inadequacies can lead to discontinuity of care, delayed diagnoses, polypharmacy, weak follow-up plans, repeated and unnecessary tests, and physicians' inability to recognize the need for referral. All of these factors can result in reduced quality of care, medical errors, and increased healthcare costs [24]. When compared to the study done in Iraq, Iran, and Saudi Arabia, this finding was better in that the quality of referral letters was poor at 69.5% 30.5%, and 37% respectively [21,22,25]. This might be due because of a difference in the study period.

**Table 6. Factors associated with the Quality of referral system towards Dessie Comprehensive Specialized Hospital, Northeast, Ethiopia, 2022/23 (n = 413).**

| Variable | Patient Satisfaction | | COR (95% CI) | AOR (95% CI) | P value |
|---|---|---|---|---|---|
| | Good | Poor | | | |
| **Sex** | | | | | |
| Male | 83 | 118 | 0.773(0.524-1.141) | 0.989(0.602-1.63) | 0.966 |
| Female | 101 | 111 | 1 | 1 | |
| **Religion** | | | | | 0.261 |
| Orthodox | 89 | 107 | 0.475(.135-1.676) | 0.42(0.096-1.80) | 0.241 |
| Muslim | 80 | 103 | 0.444(0.126-1.569) | 0.30(0.07-1.32) | 0.111 |
| Protestant | 8 | 15 | 0.305(0.068-1.364) | 0.28(0.05-1.57) | 0.148 |
| Catholic | 7 | 4 | 1 | 1 | |
| **Marital status** | | | | | 0.456 |
| Single | 36 | 49 | 1.469(0.614-3.516) | 1.76(0.63-4.94) | 0.285 |
| Married | 124 | 137 | 1.810(0.816-4.017) | 1.91(0.77-4.78) | 0.165 |
| Divorced | 14 | 23 | 1.217(0.444-3.338) | 1.25(0.40-3.86) | 0.701 |
| Widowed | 10 | 20 | 1 | 1 | |
| **Educational status** | | | | | 0.970 |
| Unable to read & write | 14 | 26 | 0.588(0.279-1.242) | 0.91(0.39-2.13) | 0.830 |
| Able to read & write | 32 | 40 | 0.874(0.483-1.582) | 1.06(0.54-2.09) | 0.864 |
| Primary education | 40 | 49 | 0.892(0.511-1.56) | 0.84(0.45-1.58) | 0.593 |
| Secondary education | 44 | 55 | 0.874(0.509-1.502) | 0.90(0.49-1.64) | 0.721 |
| College & above | 54 | 59 | 1 | 1 | |
| **Occupation** | | | | | 0.043 |
| Farmer | 27 | 48 | 1 | 1 | |
| Housewife | 40 | 33 | 2.155(1.114-4.167) | 1.56(0.71-3.40) | 0.266 |
| Merchant | 33 | 36 | 1.630(0.836-3.176) | 1.20(0.54-2.63) | 0.656 |
| Government Employee | 51 | 78 | 1.162(0.645-2.095) | 0.60(0.29-1.24) | 0.165 |
| Daily labor | 13 | 14 | 1.651(0.678-4.020) | 1.12(0.39-3.19) | 0.835 |
| Other(student/drivers) | 20 | 20 | 1.778(0.816-3.873) | 2.14(0.75-6.16) | 0.157 |
| **Residency** | | | | | |
| Rural | 48 | 101 | 0.447(0.294-0.681) | **0.33(0.19−.56)** | **0.000** |
| Urban | 136 | 128 | 1 | 1 | |
| **Family size** | | | | | |
| <5 members | 124 | 155 | 0.987(0.652-1.493) | 0.84(0.49-1.42) | 0.509 |
| ≥5 Members | 60 | 74 | 1 | 1 | |
| **Mode of transportation** | | | | | **0.000** |
| Ambulance | 135 | 112 | 5.183(2.999-8.96) | **6.59(3.49-12.43)** | **0.000** |
| Private vehicle | 20 | 19 | 4.526(2.05-10.02) | **5.62(2.33-10.52)** | **0.000** |
| On foot | 20 | 86 | 1 | **1** | |
| Other(Public taxi) | 9 | 12 | 3.23(1.196-8.70) | **3.82(1.25-8.65)** | **0.019** |
| **Type of cases** | | | | | 0.391 |
| Accident | 34 | 44 | 1.388(0.795-2.424) | 0.84(0.44-1.62) | 0.610 |
| Medical | 54 | 97 | 1 | 1 | |
| Surgical | 29 | 35 | 1.488(0.822-2.696) | 1.16(0.59-2.27) | 0.673 |
| MCH | 60 | 49 | 2.200(1.330-3.638) | 1.53(0.84-2.78) | 0.165 |
| Others | 7 | 4 | 3.144(0.88-11.224) | 2.16(0.55-8.51) | 0.272 |

In this study, patients were referred somewhat incompletely, with referral papers that were often inadequate. However, since a referral paper serves as a channel for transferring information, it is essential for effective communication between the referring and receiving health professionals. Insufficient information can negatively impact patients' health outcomes and the health system as a whole. This discrepancy may be due to differences in organizational management policies, the status of health professionals, their knowledge and practices regarding the referral system, and the level of health facilities [9,26].

The overall quality of the referral feedback paper was good in 362 (87.7%) cases (CI 84.5%–90.6%). This was better compared to the studies conducted in Iraq and Saudi Arabia, where the findings were 78.51% and 39%, respectively [18,21,22]. The difference may be due to variations in socioeconomic status and differences among health professionals (such as number, knowledge, and practice) regarding referrals. The overall quality of the referral process was good in 365 cases (88.3%) (CI 86.5%–91.6%). When compared to studies conducted in Iraq, Iran, and Saudi Arabia, this finding was better, as the quality of referral letters and feedback was poor in 69.5%, 30.5%, and 37%, respectively. [21,22,25]. The difference may be due to variations in socioeconomic status and differences among health professionals (such as number, knowledge, and practice) regarding referrals. The overall quality of the referral process was good in 365 cases (88.3%) (CI 86.5%–91.6%). When compared to studies conducted in Iraq, Iran, and Saudi Arabia, this finding was better, as the quality of referral letters and feedback was poor in 69.5%, 30.5%, and 37%, respectively.

Concerning patient satisfaction, our findings revealed that 44.6% (CI 40.00%−49.20%) of the referred patients were satisfied with the referral system of health facilities. This finding was in line with a study conducted in Eastern Tigray, which reported a satisfaction rate of 47%. [18] lower than a study conducted in Addis Ababa, Ethiopia 73.5%, CI (67.9%–81.1%) [27]. This discrepancy might be related to differences in the health infrastructure of Addis Ababa.

Overall, the quality of the referral system to Dessie Comprehensive Specialized Hospital, based on the three components of health facility referral quality measures, was 62.5% (range: 57.6%–66.8%). This is higher compared to studies conducted in Nigeria and Debre Birhan, which reported rates of 53.2% and 47.5%, respectively. This difference may be due to variations in the socio-demographic and cultural characteristics of the respondents [6,28]. This could be also associated with a lack of appropriate feedback, a high burden of tasks, incorrect referrals, and poor referral communication [5,10].

In this study, mode of transportation and residency were statistically associated with patient satisfaction regarding referrals to health facilities. Access to transportation by ambulance, private vehicle, or public taxi service was positively associated with patient satisfaction with referrals to health facilities. These findings are consistent with studies conducted in America, China, and developing countries, including Malawi [29–31]. In comparison to a study conducted in developing countries, including Malawi, using a qualitative approach and narrative reviews, similarly, those who used ambulances were positively associated with the quality of referral [5,32,33]. The possible justification for this finding could be that having access to transportation saves time and costs, thereby facilitating easier reaching of the referral site and enabling early initiation of the needed health care services. On the other hand, residing in an urban area was positively associated with patient satisfaction regarding referrals to health facilities. This finding is similar to a study conducted in Tigray, Northern Ethiopia. This may be related to the fact that rural residents often face challenges such as limited transportation access, long travel times to reach the referral destination, higher transportation costs, and increased time required to receive services [18].

## Strengths and limitations of the study

The study design was cross-sectional, which does not establish cause-and-effect relationships. Additionally, since the study was conducted at a health facility (institution), there may be social desirability bias. Nonetheless, it provided valuable insights into the quality of the referral system and associated factors in the study area..

## Conclusion

Generally, while the referral system at Dessie Comprehensive Specialized Hospital demonstrated strong performance in its structural inputs (90.3%) and the quality of its referral and feedback documentation processes (overall process quality of 88.3%), overall patient satisfaction with the referral service was notably low at 44.6%. This low patient satisfaction significantly contributed to an overall calculated referral quality of 62.5%, indicating that a substantial proportion of referred clients experienced suboptimal referral quality. The referring health facilities face challenges, as some lack necessary inputs for effective referrals—such as a focal person for coordination—and many need to improve their feedback systems.

It is recommended that the Amhara Health Bureau arrange and facilitate a transportation mechanism, particularly ambulances within the region, and establish an effective feedback system. Dessie Comprehensive Specialized Hospital should ensure that feedback is sent back to the referring health facilities. All health facilities should prioritize improving client satisfaction and ensuring the completeness of the referral documents they use. They should also facilitate communication with referral centers to receive feedback for continuous improvement. Finally, health professionals should ensure that referral and feedback documents are completed accurately and properly.

## Supporting information

**S1 File. "Combined figures and table showing overall quality towards referral among referral cases, socio-demographic characteristics, frequency distribution of components of the referral papers among referral cases, frequency distribution of components of feedback paper among referral cases, assessment of referral and feedback paper quality among referral cases, patient satisfaction toward referral among referral cases and factors associated with the quality of referral system.**
(CSV)

## Acknowledgments

We would like to thank Wollo University and Dessie Comprehensive Specialized Hospital administrators and staff for their cooperation.

## Author contributions

**Conceptualization:** Biruk Abera, Toyib Yasin, Hiwot Gizaw.

**Data curation:** Biruk Abera, Toyib Yasin, Hiwot Gizaw.

**Formal analysis:** Biruk Abera, Toyib Yasin, Hiwot Gizaw.

**Funding acquisition:** Biruk Abera, Toyib Yasin, Hiwot Gizaw.

**Investigation:** Biruk Abera, Hiwot Gizaw.

**Methodology:** Yonas Fissha Adem, Biruk Abera.

**Project administration:** Biruk Abera.

**Resources:** Biruk Abera.

**Software:** Biruk Abera, Toyib Yasin.

**Validation:** Biruk Abera.

**Visualization:** Biruk Abera.

**Writing – original draft:** Yonas Fissha Adem, Biruk Abera.

**Writing – review & editing:** Yonas Fissha Adem, Biruk Abera.

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
