## [Decision Letter · Decision Letter 0]

23 Oct 2024

PLOS ONE

Dear Dr. Adem,

Thank you for submitting your manuscript to PLOS ONE. After careful consideration, we feel that it has merit but does not fully meet PLOS ONE’s publication criteria as it currently stands. Therefore, we invite you to submit a revised version of the manuscript that addresses the points raised during the review process.

Thank you for your submission. Could you please ensure that your manuscript follows the STROBE checklist for cross-sectional studies and adheres to the PLOS ONE guidelines for figures and tables? Additionally, we kindly ask that you provide more details on how reliability and validity were ensured in your study.

Best regards

We look forward to receiving your revised manuscript.

Kind regards,

Trhas Tadesse Berhe, PhD

Academic Editor

PLOS ONE

Journal Requirements: When submitting your revision, we need you to address these additional requirements. 1. Please ensure that your manuscript meets PLOS ONE's style requirements, including those for file naming. The PLOS ONE style templates can be found at https://journals.plos.org/plosone/s/file?id=wjVg/PLOSOne_formatting_sample_main_body.pdf and https://journals.plos.org/plosone/s/file?id=ba62/PLOSOne_formatting_sample_title_authors_affiliations.pdf 2. When completing the data availability statement of the submission form, you indicated that you will make your data available on acceptance. We strongly recommend all authors decide on a data sharing plan before acceptance, as the process can be lengthy and hold up publication timelines. Please note that, though access restrictions are acceptable now, your entire data will need to be made freely accessible if your manuscript is accepted for publication. This policy applies to all data except where public deposition would breach compliance with the protocol approved by your research ethics board. If you are unable to adhere to our open data policy, please kindly revise your statement to explain your reasoning and we will seek the editor's input on an exemption. Please be assured that, once you have provided your new statement, the assessment of your exemption will not hold up the peer review process. 3. In the online submission form, you indicated that "The datasets generated and analyzed during the current study are not publicly available due to the risks in identifying participants as true anonymization would be difficult to guarantee, but subsets of the data can be available from the corresponding author upon a reasonable request." All PLOS journals now require all data underlying the findings described in their manuscript to be freely available to other researchers, either 1. In a public repository, 2. Within the manuscript itself, or 3. Uploaded as supplementary information.This policy applies to all data except where public deposition would breach compliance with the protocol approved by your research ethics board. If your data cannot be made publicly available for ethical or legal reasons (e.g., public availability would compromise patient privacy), please explain your reasons on resubmission and your exemption request will be escalated for approval. 4. Please include a separate caption for each figure in your manuscript. 5. We note you have included a table to which you do not refer in the text of your manuscript. Please ensure that you refer to Table 6 in your text; if accepted, production will need this reference to link the reader to the Table. 6. Please include a copy of Table 7 which you refer to in your text on page 28 in PDF submission. 7. Please upload a copy of Supporting Information Figure/Table/etc. Supporting Information which you refer to in your text on page 41 in PDF submission.

Reviewers' comments:

Reviewer's Responses to Questions

**Comments to the Author**

1. Is the manuscript technically sound, and do the data support the conclusions?

Reviewer #1: Partly

Reviewer #2: Yes

2. Has the statistical analysis been performed appropriately and rigorously?

Reviewer #1: Yes

Reviewer #2: Yes

3. Have the authors made all data underlying the findings in their manuscript fully available?

Reviewer #1: Yes

Reviewer #2: Yes

4. Is the manuscript presented in an intelligible fashion and written in standard English?

Reviewer #1: Yes

Reviewer #2: Yes

Reviewer #1: Dear author(s), thank you for your manuscript.

Overall, this is a clear, concise, and well-written manuscript on an important topic that is often under-researched. The introduction is relevant. Sufficient information about the importance of the research topic is presented for readers to follow the present study rationale and procedures. Although the article is well written but clarification of a few details should be provided.

Specific comments follow:

1. What was the reason for choosing the study method? As the study uses a cross-sectional design, Can this method establish cause and effect relationships between variables?

2. What was the reason for choosing only one hospital? The study only includes patients referred to one hospital. This geographic limitation means the findings may not apply to other hospitals with different patient demographics or healthcare infrastructures.

3. The article predominantly uses descriptive statistics to report results (e.g., frequencies and percentages), but lacks deeper inferential statistical analyses to explore more complex relationships between variables. This restricts the depth of insight that can be gained from the data.

4. While some operational definitions are provided, others like “good” and “poor” referral quality are based on arbitrary cutoffs (75%), without a robust explanation of why these thresholds were chosen. This raises questions about the validity and reliability of the quality assessments.

5. Since data collection was done in a healthcare facility, patients might have provided more favorable responses due to social desirability bias, particularly in reporting their satisfaction. This bias could skew the findings, but the authors do not discuss how they managed or mitigated this issue.

6. The article mentions that verbal informed consent was obtained, but it does not explain why written consent was not sought or how verbal consent was documented, raising concerns about ethical rigor. Additionally, it does not address any potential risks to participants or how these were mitigated.

7. Given the global trend towards electronic referral systems, the absence of any discussion on the potential role of digital solutions (e.g., e-referrals, telemedicine) in improving referral quality is a missed opportunity, especially in regions with infrastructure development plans like Ethiopia.

Reviewer #2: The main text mentions Table 7, but the table does not actually exist. Please recheck the arrangement and ensure that all referenced tables are correctly accounted for.

The resolution of the figures should be improved. Please resubmit new, clearer figures to ensure better publication quality.

Conclusion subtitle has a unexpected dot (line 412)

**Do you want your identity to be public for this peer review?** For information about this choice, including consent withdrawal, please see our Privacy Policy

Reviewer #1: No

Reviewer #2: No

---

## [Author Response · Author response to Decision Letter 1]

6 Dec 2024

Date: 01 /11/2024

Trhas Tadesse Berhe

PLOS ONE

Dear… D.r Trhas Tadesse Berhe

Thank you for giving us the opportunity to submit a revised draft of our manuscript titled

“Quality of Referral System and Associated Factors Among Referred Clients Referred to Dessie Comprehensive Specialized Hospital, Northeast, Ethiopia” to the PLOS ONE. We appreciate the time and effort that you and the reviewers have dedicated to providing your valuable feedback on our manuscript. We are grateful to the reviewers for their insightful comments on our paper. We have been able to incorporate changes to reflect most of the suggestions provided by the reviewers. We have highlighted the changes within the manuscript.

Please let us know if you still have any questions or concerns about the manuscript. We will be happy to address them, now in a timely manner.

Sincerely,

Yonas Fissha Adem

Reviewer 1

Reviewer #1: Dear author(s), thank you for your manuscript.

Overall, this is a clear, concise, and well-written manuscript on an important topic that is often under-researched. The introduction is relevant. Sufficient information about the importance of the research topic is presented for readers to follow the present study rationale and procedures. Although the article is well written but clarification of a few details should be provided.

Response: dear reviewer, thank you very much for your motivational comment.

Specific comments follow:

1. What was the reason for choosing the study method? As the study uses a cross-sectional design, Can this method establishes cause and effect relationships between variables?

Response: dear reviewer, thank you for your valuable feedback. a cross-sectional design doesn’t show cause and effect relationships between variables because it is collected data at a single point in a time. But before now there is not studies conduct in Ethiopian especially in the study areas so it better to know first prevalence of good quality of referral system. So, to achieved this purpose cross-sectional study was preferred to conduct.

2. What was the reason for choosing only one hospital? The study only includes patients referred to one hospital. This geographic limitation means the findings may not apply to other hospitals with different patient demographics or healthcare infrastructures.

Response: dear reviewer, thank you for spotting this. We are choosing this hospital because it’s the only Comprehensive Specialized Hospital in wollo, which currently provides services for 10 million people in the South Wollo, North Wollo, Waghimra, and Oromia specialized zones.

3. The article predominantly uses descriptive statistics to report results (e.g., frequencies and percentages), but lacks deeper inferential statistical analyses to explore more complex relationships between variables. This restricts the depth of insight that can be gained from the data.

Response: Thank you for your thoughtful concern of our study. But this study was done both descriptive and analytical study. Descriptive study consists frequencies and percentages in this study. Analytical study consists comparison between dependent and independent variables using bivariable and multivariable binary logistic regression analysis in this study. So we consider this study have deeper inferential statistical analyses.

4. While some operational definitions are provided, others like “good” and “poor” referral quality are based on arbitrary cutoffs (75%), without a robust explanation of why these thresholds were chosen. This raises questions about the validity and reliability of the quality assessments.

Response: Thank you for your detailed comments. We are using “good” and “poor” referral quality system based on reference from previous studies. It isn’t arbitrary cutoffs. We have already putted a reference.

5. Since data collection was done in a healthcare facility, patients might have provided more favorable responses due to social desirability bias, particularly in reporting their satisfaction. This bias could skew the findings, but the authors do not discuss how they managed or mitigated this issue.

Response: Thank you so much for your careful check. But we have deployed four BSc. Nurses data collector and two BSc Nurses supervisor from other health facilities to reduce social desirability bias. The data collector informed the study participants that they came from other health facilities and its objective of the studies.

6. The article mentions that verbal informed consent was obtained, but it does not explain why written consent was not sought or how verbal consent was documented, raising concerns about ethical rigor. Additionally, it does not address any potential risks to participants or how these were mitigated.

Response: Thank you for your valuable input. Verbal consent is appropriate for routine procedures that don't pose a significant risk to the patient and no need of written informed consent because we didn’t do any invasive procedures here rather we simply collected data from respondents. But we have incorporated the issue about mitigate potential risks to participants.

7. Given the global trend towards electronic referral systems, the absence of any discussion on the potential role of digital solutions (e.g., e-referrals, telemedicine) in improving referral quality is a missed opportunity, especially in regions with infrastructure development plans like Ethiopia.

Response: Thank you for raising this important point. We have incorporated the issue you raised in the introduction part of the revised manuscript according to your comment.

Reviewer #2: The main text mentions Table 7, but the table does not actually exist. Please recheck the arrangement and ensure that all referenced tables are correctly accounted for.

The resolution of the figures should be improved. Please resubmit new, clearer figures to ensure better publication quality.

Response: We would like to thank the dear reviewer for your frank comment. We reviewed and made adjustments in the revised manuscript.

Conclusion subtitle has a unexpected dot (line 412)

Response: We appreciate your valuable suggestion. We have made corrected in the revised manuscript as per your expectations.

---

## [Decision Letter · Decision Letter 1]

15 Apr 2025

Dear Dr. Yonas Fissha Adem,

Thank you for submitting your manuscript to PLOS ONE. After careful consideration, we feel that it has merit but does not fully meet PLOS ONE’s publication criteria as it currently stands. Therefore, we invite you to submit a revised version of the manuscript that addresses the points raised during the review process.

Please ensure that the reviewers' comments are fully addressed, particularly those related to the sampling procedure, the clarity and coherence of the discussion section, and the alignment of the conclusion with the study findings. Additionally, we recommend that the manuscript be thoroughly proofread by a professional language editor prior to submission to improve overall clarity and academic tone

We look forward to receiving your revised manuscript.

Kind regards,

Trhas Tadesse Berhe, PhD

Academic Editor

PLOS ONE

Reviewers' comments:

Reviewer's Responses to Questions

**Comments to the Author**

Reviewer #1: All comments have been addressed

Reviewer #3: (No Response)

2. Is the manuscript technically sound, and do the data support the conclusions?

Reviewer #1: Yes

Reviewer #3: Partly

3. Has the statistical analysis been performed appropriately and rigorously?

Reviewer #1: I Don't Know

Reviewer #3: I Don't Know

4. Have the authors made all data underlying the findings in their manuscript fully available?

Reviewer #1: Yes

Reviewer #3: Yes

5. Is the manuscript presented in an intelligible fashion and written in standard English?

Reviewer #1: Yes

Reviewer #3: Yes

Reviewer #1: (No Response)

Reviewer #3: Thank you for the opportunity to review this manuscript. It is a worthwhile study of an interesting subject, however there are a number of things that should be addressed before it is published.

The first 2 sentences of the abstract are not relevant to the study and should be deleted.

The components assessed to determine the overall quality, input quality and process quality of the referral are not provided until the Tables in the body of the results. It would be helpful to have some examples in the Abstract and Methods eg Process Quality measured whether the important patient demographic and clinical information was included in the referral, Outcome quality was based on the patient’s impression of their referral etc.

It is not clear what “focal person” means. Does it mean ‘contact person’? The presence or absence of a “focal person” is mentioned in the introduction and discussion, but not in the methods or results so it is not clear how this was measured or what was found. Either the methods and results should be included, or the concept removed from the introduction and discussion.

The recruitment strategy and numbers do not seem realistic and require clarification. The methods state that every 4th patient of an expected 2000 patients per month was approached and interviewed. It is inconceivable that there were no protocol violations during this process. What happened if patients presented overnight, if multiple patients presented at once, if patients were too confused or unwell or not available to respond to questions, if the research staff were sick? The actual recruitment strategy should be explained. If it is a convenience sample of ~1/4 of patients presenting to the hospital, that should be stated.

Reference 16 reports that in Tigray, 82% of patients were self-referred. Dessie Comprehensive Specialized Hospital presumably treats similar numbers of self-referred patients. Are the 2000 patients per month only the ‘referred’ patients, (ie does the hospital see 10000 patients per month, of whom 20% are referred) or is this all the patients, and what proportion were referred? How did the research staff distinguish between self-referred patients and those transferred from other hospitals on foot or by public taxi, some of whom are likely to have lost their referral papers? Inclusion or exclusion of patients with missing papers could significantly change the results.

Patient satisfaction is reported as 44.6% overall, but the individual questions suggest a much higher level of satisfaction. How was the 44.6% overall satisfaction derived? Similarly, the overall quality of the referral system is reported as 62.5%, but in the data tables 94.7% of the referrals were good, and 87.3% of the feedback was good. Can this please be clarified? Similarly, the conclusions that’ generally clients had poor referral quality’ are not supported by the data, which shows that most referrals were good quality.

The paper would benefit from editing to improve the academic English.

**Do you want your identity to be public for this peer review?** For information about this choice, including consent withdrawal, please see our Privacy Policy

Reviewer #1: No

Reviewer #3: No

---

## [Author Response · Author response to Decision Letter 2]

4 Jul 2025

Date: 01 /06/2025

Trhas Tadesse Berhe

PLOS ONE

Dear… D.r Trhas Tadesse Berhe

Thank you for giving us the opportunity to submit a revised draft of our manuscript titled

“Quality of Referral System and Associated Factors Among Referred Clients Referred to Dessie Comprehensive Specialized Hospital, Northeast, Ethiopia” to the PLOS ONE. We appreciate the time and effort that you and the reviewers have dedicated to providing your valuable feedback on our manuscript. We are grateful to the reviewers for their insightful comments on our paper. We have been able to incorporate changes to reflect most of the suggestions provided by the reviewers. We have highlighted the changes within the manuscript.

Please let us know if you still have any questions or concerns about the manuscript. We will be happy to address them, now in a timely manner.

Sincerely,

Yonas Fissha Adem

Point by Point Response to - Editor

Point by Point Response to – Reviewer

Reviewer #3: Thank you for the opportunity to review this manuscript. It is a worthwhile study of an interesting subject; however there are a number of things that should be addressed before it is published.

The first 2 sentences of the abstract are not relevant to the study and should be deleted.

Response: We appreciate your thoughtful observation regarding this issue. We've made changes to this section based on your comments.

The components assessed to determine the overall quality, input quality and process quality of the referral are not provided until the Tables in the body of the results. It would be helpful to have some examples in the Abstract and Methods eg Process Quality measured whether the important patient demographic and clinical information was included in the referral, Outcome quality was based on the patient’s impression of their referral etc.

Response: Thank you for your suggestion, dear. However, in this study, the components of the quality referral system (structure, process, and outcome) have already been mentioned in the operational definition section of the methods. We can explain their meanings in the abstract, but to keep it short and precise; we haven’t mentioned them in the abstract.

It is not clear what “focal person” means. Does it mean ‘contact person’? The presence or absence of a “focal person” is mentioned in the introduction and discussion, but not in the methods or results so it is not clear how this was measured or what was found. Either the methods and results should be included, or the concept removed from the introduction and discussion.

Response: We appreciate your thoughtful observation regarding this issue. But, We've already incorporated about “focal person” in the Method part “operational definition” and in result part “Structural (input) part of the referral system”. "Focal person" typically refers to an individual designated to oversee and coordinate activities related to a specific program, initiative, or area of focus within the hospital.

The recruitment strategy and numbers do not seem realistic and require clarification. The methods state that every 4th patient of an expected 2000 patients per month was approached and interviewed. It is inconceivable that there were no protocol violations during this process. What happened if patients presented overnight, if multiple patients presented at once, if patients were too confused or unwell or not available to respond to questions, if the research staff were sick? The actual recruitment strategy should be explained. If it is a convenience sample of ~1/4 of patients presenting to the hospital, that should be stated.

Response: Thank you for your suggestion, dear. However, this study's recruitment strategy was systematic random sampling, selecting every 4th patient out of an expected 2,007 patients monthly. Multiple patients cannot be presented at once because their charts are organized based on their assigned order. Therefore, every 4th patient was selected; patients can come during the night, but patient chart was given based on the order so no problem for random patient selection, but the data collection was conducted in day. If a patient was confused or unable to communicate, the next random sample was taken. No research staffs were sick during our study, but if any staff member had fallen ill, other data collectors would have been deployed.

Reference 16 reports that in Tigray, 82% of patients were self-referred. Dessie Comprehensive Specialized Hospital presumably treats similar numbers of self-referred patients. Are the 2000 patients per month only the ‘referred’ patients, (ie does the hospital see 10000 patients per month, of whom 20% are referred) or is this all the patients, and what proportion were referred? How did the research staff distinguish between self-referred patients and those transferred from other hospitals on foot or by public taxi, some of whom are likely to have lost their referral papers? Inclusion or exclusion of patients with missing papers could significantly change the results.

Response: Thank you for your suggestion dear, but this study is aimed to assess the quality of referral system but not about magnitude of self-referred patients. Every patient should bring referral paper so if it’s lost it should be rewrite again to get treatment from Hospital since its referral hospital. But in our stud, no patients loss the referral paper. So no need of putting this idea in the inclusion and exclusion criteria.

Patient satisfaction is reported as 44.6% overall, but the individual questions suggest a much higher level of satisfaction. How was the 44.6% overall satisfaction derived? Similarly, the overall quality of the referral system is reported as 62.5%, but in the data tables 94.7% of the referrals were good, and 87.3% of the feedback was good. Can this please be clarified? Similarly, the conclusions that’ generally clients had poor referral quality’ are not supported by the data, which shows that most referrals were good quality.

Response: Thank you for your suggestions. In this stud, the overall patient satisfaction with the referral system of health facilities to Dessie Comprehensive Specialized Hospital was found to be 44.6%, this result was found by calculating the average value of each 8 measuring variables of patient satisfaction for each individual then patients who scored above the half(50%) was considered satisfied. Even when you see each individual variables value, it’s not highest and it was classified into five. In addition to this the overall quality of referral system was computed by three components of quality. So even if the structural and process parts of the referral system have highest score which was 90.3% and 94.7% respectively. But the outcome (patient satisfaction) was poor which score 44.6%. So the overall quality of referral was calculated by computing the above three referral system so final we have got 62.5% of the patients scored 75 % and above.

The paper would benefit from editing to improve the academic English.

Response: Thank you for your suggestions. We will improve the whole document based on your comments.

---

## [Editor Report · Decision Letter 2]

10 Jul 2025

Thank you for submitting your manuscript to PLOS ONE. After careful consideration, we feel that it has merit but does not fully meet PLOS ONE’s publication criteria as it currently stands. Therefore, we invite you to submit a revised version of the manuscript that addresses the points raised during the review process.

Thank you for your response. However, the explanation regarding the feasibility of systematic sampling and the absence of any protocol violations appears unrealistic. Please revise your response and the Methods section to acknowledge potential real-world challenges (e.g., patient unavailability, staff absence, night-time arrivals), and explain how such issues were handled or mitigated. Transparency about possible limitations is important for methodological rigor.

-the reviewer’s request was not fully addressed. While the components of quality (structure, process, and outcome) are defined in the operational definitions, the reviewer specifically asked for brief examples to be included in the Abstract and Methods sections for clarity.

Please revise the Abstract to include a short example for each component, and add a concise explanation in the Methods section to improve reader understanding

-While the term “focal person” is defined in the operational definitions and mentioned in the results, it would benefit readers if it were clearly introduced and defined upon first use in the Introduction.

Please revise the manuscript to define the term earlier in the narrative to improve clarity and flow.

-the reviewer’s concern was not fully addressed. While we understand the focus was not on self-referral, the issue raised relates to potential sample bias due to possible exclusion of patients who may have lost referral papers.

Please clarify how such cases were handled and revise the inclusion criteria and limitations sections to reflect this, ensuring transparency in your methodology.

-The explanation regarding the overall satisfaction score and referral quality remains unclear. There appears to be a misalignment between the findings and the conclusion, particularly as structure and process scores were high, yet the conclusion states that clients experienced “poor referral quality.”

Please provide a clearer explanation of how the overall quality score was calculated, especially how satisfaction was incorporated. Also, revise the conclusion to more accurately reflect the mixed findings  for example, acknowledging high structural and process performance alongside lower patient satisfaction.

-Thank you for acknowledging the need to improve the language quality of the manuscript. However, your response does not indicate whether a thorough language revision has been carried out.

To ensure clarity and academic quality, please revise the manuscript with support from a native English speaker or a professional language editor, and submit a track-changed version highlighting the edits made.

We look forward to receiving your revised manuscript.

Kind regards,

Trhas Tadesse Berhe, PhD

Academic Editor

PLOS ONE
---

## [Author Response · Author response to Decision Letter 3]

2 Sep 2025

Date: 20 /08/2025

Trhas Tadesse Berhe

PLOS ONE

Dear… D.r Trhas Tadesse Berhe

Thank you for giving us the opportunity to submit a revised draft of our manuscript titled

“Quality of Referral System and Associated Factors Among Referred Clients Referred to Dessie Comprehensive Specialized Hospital, Northeast, Ethiopia” to the PLOS ONE. We appreciate the time and effort that you and the reviewers have dedicated to providing your valuable feedback on our manuscript. We are grateful to the reviewers for their insightful comments on our paper. We have been able to incorporate changes to reflect most of the suggestions provided by the reviewers. We have highlighted the changes within the manuscript.

Please let us know if you still have any questions or concerns about the manuscript. We will be happy to address them, now in a timely manner.

Sincerely,

Yonas Fissha Adem

Point by Point Response to - Editor

Point by Point Response to – Reviewer

Thank you for your response. However, the explanation regarding the feasibility of systematic sampling and the absence of any protocol violations appears unrealistic. Please revise your response and the Methods section to acknowledge potential real-world challenges (e.g., patient unavailability, staff absence, night-time arrivals), and explain how such issues were handled or mitigated. Transparency about possible limitations is important for methodological rigor.

Response: Response: We appreciate your thoughtful observation regarding this issue. We've made changes to this section based on your comments.

-the reviewer’s request was not fully addressed. While the components of quality (structure, process, and outcome) are defined in the operational definitions, the reviewer specifically asked for brief examples to be included in the Abstract and Methods sections for clarity.Please revise the Abstract to include a short example for each component, and add a concise explanation in the Methods section to improve reader understanding

Response: Thank you for your suggestions. We've made incorporated to the Abstract and Methods section based on your comments.

-While the term “focal person” is defined in the operational definitions and mentioned in the results, it would benefit readers if it were clearly introduced and defined upon first use in the Introduction. Please revise the manuscript to define the term earlier in the narrative to improve clarity and flow.

Response: Thank you for your suggestions. We have defined the raised issue in the introduction part.

-the reviewer’s concern was not fully addressed. While we understand the focus was not on self-referral, the issue raised relates to potential sample bias due to possible exclusion of patients who may have lost referral papers.

Please clarify how such cases were handled and revise the inclusion criteria and limitations sections to reflect this, ensuring transparency in your methodology.

Response: Thank you for your suggestions. But in our study, no patients loss the referral paper. We want to ask a great apologize for making a mistake here and we have replaced loss the referral paper with unable to communicate in the exclusion criteria.

-The explanation regarding the overall satisfaction score and referral quality remains unclear. There appears to be a misalignment between the findings and the conclusion, particularly as structure and process scores were high, yet the conclusion states that clients experienced “poor referral quality.”

Please provide a clearer explanation of how the overall quality score was calculated, especially how satisfaction was incorporated. Also, revise the conclusion to more accurately reflect the mixed findings for example, acknowledging high structural and process performance alongside lower patient satisfaction.

Response: Thank you for your suggestions. The perceived misalignment arises because the overall quality of the referral system was calculated by considering three components of quality measures: input, process, and outcome. While the structural (input) and process aspects scored highly, the outcome component, specifically patient satisfaction, was considerably lower, which pulled down the overall quality score.

• Structural (Input) Quality: The overall structural input quality of the referral system was 90.3%. This indicates a high availability of essential components like a referral focal person, registry book, standard referral paper, and transport.

• Process Quality: The overall quality related to the referral and feedback papers was also high.

◦ The overall quality of the referral paper was good in 94.7% of participants.

◦ The score of components of feedback sent was good in quality in 87.3% of study subjects, and the overall quality of the referral feedback was good in 87.7% of participants.

◦ The overall process quality was good in 88.3% of study participants.

• Outcome (Patient Satisfaction): This is where a significant difference is observed. The overall patient satisfaction with the referral system at Dessie Comprehensive Specialized Hospital was only 44.6% (95% CI: 40.00%–49.20%). For example, only 68.3% of referred patients agreed that health professionals explained the referral in a way they could understand, and 62.5% agreed that professionals included them in deciding the reason for the referral.

• Overall Referral Quality: When all three components (input, process, and outcome) were considered, the overall quality of the referral system was computed to be 62.5% (95% CI 57.6%-66.8%), meaning 37.5% was considered poor in quality.

• The conclusion part has accordingly modified

-Thank you for acknowledging the need to improve the language quality of the manuscript. However, your response does not indicate whether a thorough language revision has been carried out.

To ensure clarity and academic quality, please revise the manuscript with support from a native English speaker or a professional language editor, and submit a track-changed version highlighting the edits made.

Response: Thank you for your suggestions. We have improved as much as possible. But we didn’t highlight the change because many grammatical and spelling correction might presented.

---

## [Editor Report · Decision Letter 3]

22 Sep 2025

Dear Dr. Adem,

Thank you for submitting your manuscript to PLOS ONE. After careful consideration, we feel that it has merit but does not fully meet PLOS ONE’s publication criteria as it currently stands. Therefore, we invite you to submit a revised version of the manuscript that addresses the points raised during the review process.

I  appreciate the efforts you have made in addressing the reviewer’s comments. However, after careful review, it is evident that several key concerns raised by the reviewers have not been fully resolved. To move your manuscript forward in the review process, we kindly ask you to provide further revisions as outlined below:

**Systematic Sampling & Real-World Challenges**While your revision briefly mentions night-time arrivals and uncommunicative patients, the explanation remains limited. Kindly expand the *Methods*  section to explicitly acknowledge common real-world challenges (e.g., patient unavailability, staff absence, timing of arrivals) and describe how these were handled in practice.**Examples of Quality Components (Abstract and Methods)**In the *Methods*  section, you have outlined structure, process, and outcome components. However, the *Abstract*  still does not include short illustrative examples (e.g., structure = referral forms/focal person, process = referral and feedback papers, outcome = patient satisfaction). Kidly  revise the *Abstract*  accordingly for clarity.**Sample Bias  Referral Papers**The reviewer raised concerns about potential bias if patients lost referral papers. Your current response simply replaces this with “unable to communicate” in the exclusion criteria. Please clarify whether any such cases occurred, how they were handled, and acknowledge this as a possible limitation if applicable.**Language Revision**Although you mention improving the manuscript, there is no evidence that a full language revision was carried out. Given the number of grammatical and stylistic issues still present, we strongly recommend that the manuscript be thoroughly revised by a native English speaker or a professional editor before resubmission.

I encourage you to carefully revise your manuscript in light of these points. Once the above issues are addressed, we will be happy to proceed with the next stage of review.

We look forward to receiving your revised manuscript.

Kind regards,

Trhas Tadesse Berhe, PhD

Academic Editor

PLOS ONE
---

## [Author Response · Author response to Decision Letter 4]

2 Oct 2025

Date: 01 /10/2025

Trhas Tadesse Berhe

PLOS ONE

Dear… D.r Trhas Tadesse Berhe

Thank you for giving us the opportunity to submit a revised draft of our manuscript titled

“Quality of Referral System and Associated Factors Among Referred Clients Referred to Dessie Comprehensive Specialized Hospital, Northeast, Ethiopia” to the PLOS ONE. We appreciate the time and effort that you and the reviewers have dedicated to providing your valuable feedback on our manuscript. We are grateful to the reviewers for their insightful comments on our paper. We have been able to incorporate changes to reflect most of the suggestions provided by the reviewers. We have highlighted the changes within the manuscript.

Please let us know if you still have any questions or concerns about the manuscript. We will be happy to address them, now in a timely manner.

Sincerely,

Yonas Fissha Adem

Point by Point Response to - Editor

Point by Point Response to – Reviewer

Systematic Sampling & Real-World Challenges

while your revision briefly mentions night-time arrivals and uncommunicative patients, the explanation remains limited. Kindly expand the Methods section to explicitly acknowledge common real-world challenges (e.g., patient unavailability, staff absence, timing of arrivals) and describe how these were handled in practice.

Response: We appreciate your thoughtful observation regarding this issue. We've made changes to this section based on your comments in the sampling procedures and technique.

Examples of Quality Components (Abstract and Methods)

In the Methods section, you have outlined structure, process, and outcome components. However, the Abstract still does not include short illustrative examples (e.g., structure = referral forms/focal person, process = referral and feedback papers, outcome = patient satisfaction). Kindly revise the Abstract accordingly for clarity.

Response: Thank you for your suggestions. We've made incorporated this issue in the Abstract section previously but we have put it short and precisely again based on your comments.

Sample Bias Referral Papers

The reviewer raised concerns about potential bias if patients lost referral papers. Your current response simply replaces this with “unable to communicate” in the exclusion criteria. Please clarify whether any such cases occurred, how they were handled, and acknowledge this as a possible limitation if applicable.

Response: Thank you for your suggestions. But in our study, no patient’s lost the referral paper in reality. But we have incorporated this issue in the sampling procedures and technique part based on your comments. If we have got patient lost referral paper the next random sample were taken.

Language Revision

Although you mention improving the manuscript, there is no evidence that a full language revision was carried out. Given the number of grammatical and stylistic issues still present, we strongly recommend that the manuscript be thoroughly revised by a native English speaker or a professional editor before resubmission.

Response: Thank you for your suggestions. We have improved as much as we can. But we didn’t highlight the change because many grammatical and spelling corrections might presented.

---

## [Editor Report · Decision Letter 4]

13 Nov 2025

Quality of Referral System and Associated Factors Among Referred Clients Referred to Dessie Comprehensive Specialized Hospital, Northeast, Ethiopia

PONE-D-24-24344R4

Dear Dr. Yonas Fissha Adem,

We’re pleased to inform you that your manuscript has been judged scientifically suitable for publication and will be formally accepted for publication once it meets all outstanding technical requirements.

Kind regards,

Trhas Tadesse Berhe, PhD

Academic Editor

PLOS ONE
---

## [Editor Report · Acceptance letter]

PONE-D-24-24344R4

PLOS ONE

Dear Dr. Adem,

I'm pleased to inform you that your manuscript has been deemed suitable for publication in PLOS ONE. Congratulations! Your manuscript is now being handed over to our production team.

Kind regards,

on behalf of

Dr. Trhas Tadesse Berhe

Academic Editor

PLOS ONE